# Monkeypox virus-infected individuals mount comparable humoral immune responses as Smallpox-vaccinated individuals

Ashley D. Otter [1,12] ✉, Scott Jones[1,12], Bethany Hicks [1,12], Daniel Bailey [2], Helen Callaby[2], Catherine Houlihan[2,3], Tommy Rampling[2,4,5], Nicola Claire Gordon[2], Hannah Selman[1], Panayampalli S. Satheshkumar[6], Michael Townsend[6], Ravi Mehta[7], Marcus Pond[7], Rachael Jones[8], Deborah Wright[9], Clarissa Oeser[10], Simon Tonge[11], Ezra Linley[11], Georgia Hemingway[1], Tom Coleman[1], Sebastian Millward[1], Aaron Lloyd[1], Inger Damon[6], Tim Brooks[2], Richard Vipond[9], Cathy Rowe[1] & Bassam Hallis[9]

In early 2022, a cluster of monkeypox virus (MPXV) infection (mpox) cases were identified within the UK with no prior travel history to MPXV-endemic regions. Subsequently, case numbers exceeding 80,000 were reported worldwide, primarily affecting gay, bisexual, and other men who have sex with men (GBMSM). Public health agencies worldwide have offered the IMVANEX Smallpox vaccination to these individuals at high-risk to provide protection and limit the spread of MPXV. We have developed a comprehensive array of ELISAs to study poxvirus-induced antibodies, utilising 24 MPXV and 3 Vaccinia virus (VACV) recombinant antigens. Panels of serum samples from individuals with differing Smallpox-vaccine doses and those with prior MPXV infection were tested on these assays, where we observed that one dose of Smallpox vaccination induces a low number of antibodies to a limited number of MPXV antigens but increasing with further vaccination doses. MPXV infection induced similar antibody responses to diverse poxvirus antigens observed in Smallpox-vaccinated individuals. We identify MPXV A27 as a serological marker of MPXV-infection, whilst MPXV M1 (VACV L1) is likely IMVANEX-specific. Here, we demonstrate analogous humoral antigen recognition between both MPXV-infected or Smallpox-vaccinated individuals, with binding to diverse yet core set of poxvirus antigens, providing opportunities for future vaccine (e.g., mRNA) and therapeutic (e.g., mAbs) design.

Monkeypox virus (MPXV), is a member of the Orthopoxviruses, a group of closely related viruses, some of which are highly pathogenic that cause distinctive diseases in humans and animals[1–5]. Within the Orthopoxvirus genus, other members of this viral family include Vaccinia virus (VACV), the foundation of a number of Smallpox vaccinations; Variola virus (VARV), the causative virus of Smallpox disease

which was subsequently declared eradicated in 1980[3,6]; and Cowpox virus (CPXV), the virus likely used by Edward Jenner to inoculate individuals against Smallpox[2,7]. Whilst Orthopoxviruses share a high degree of genetic homology[2,8], they vary in their pathogenicity to humans. VARV is highly pathogenic in humans, contributing to 300–500 million historic deaths worldwide, with no known animal

reservoir, aiding in its eradication. VACV can cause disease in humans[9,10], however attenuated strains of VACV have been developed through serial passage, as the basis of a number of licensed vaccines against Smallpox–Dryvax®®, ACAM2000 (both based on the New York City Board of Health vaccinia virus strain) and Modified Vaccinia Ankara (MVA) Bavarian Nordic (MVA-BN); trade-names 'IMVANEX' and 'JYNNEOS).

After the eradication of Smallpox in 1980[6], routine Smallpox vaccination, using Vaccinia virus, was halted worldwide[3,11]. Since this vaccination programme stopped, it has been suggested that the worldwide population remain increasingly at risk of poxvirus infection, due to waning antibodies in those with prior Smallpox vaccination and a lack of vaccine-derived immunity in those born after 1980[11,12].

MPXV is a zoonotic pathogen (with a presumed rodent animal reservoir) causing Mpox disease[1,13,14]. Previously, it has been identified only in central and Western African countries with occasional importations in returning travellers. There have been a small number of onward transmissions in some countries such as the UK[15–17] and USA[18–20], however minimal mutations are observed between isolates spanning multiple years[2].

In the UK on the 7th of May 2022, one case of MPXV was identified in a returning traveller from Nigeria. A week later, an autochthonous familial cluster was identified, with no link to the earlier case and no travel history outside of the UK to MPXV-endemic regions[21,22]. Shortly afterwards, further MPXV cases were identified in the UK, largely in gay, bisexual, and other men who have sex with men (GBMSM) with no travel to endemic countries or known exposure to confirmed cases. Thereafter, other non-endemic countries including the USA, Spain, Germany, Portugal and France also identified similar cases and evidence emerged of local transmission in these countries[21]. To date, >80,000 cases of MPXV have been identified globally[23], the majority of which have been in GBMSM[22]. Whole genome sequencing has identified 30–80 mutations (with the majority being nonsynonymous) in sequences from MPXV isolated during the current outbreak, which has been termed Clade IIb, compared to previous MPXV cases from Clade I or Clade IIa[24–26]. Mutations within Clade IIb were primarily either GA- to AA-, or TC- to TT substitutions, suggesting that host factors such as APOBEC3 cytosine modification led to these mutations, possibly due to sustained infection in a new host or altered phenotypic/transmission methods[25,26].

To limit transmission during the 2022 MPXV outbreak, public health agencies recommend Smallpox vaccination, as previous studies have shown protection from Mpox disease using Smallpox vaccines such as IMVANEX or ACAM2000[27–31]. The immunology of Dryvax®, ACAM2000 and IMVANEX vaccines has been well studied[27,32–36], with antibody responses detected up to 35 years post-Smallpox vaccination[34]. However, studying the immunology to MPXV infection and disease has been limited by the geographical distribution, limited number of human cases worldwide and access to convalescent serum samples. Animal models have been undertaken to understand immunology from Mpox disease in species such as macaques[28,31] and prairie dogs[27], however, the translation of this knowledge to human MPXV infections is unknown.

Whilst T-cell immune responses to MPXV infection have been studied during the current Clade IIb 2022 outbreak[37,38] and previous oubreaks[39], we sought to determine the humoral response and antigen recognition induced by both Clade IIa and IIb MPXV infection, with comparisons to IMVANEX and ACAM2000-vaccinated individuals. Similarly, using a combination of data, we demonstrate the ability of a multi-antigen ELISA to study antibody responses in Smallpox-vaccinated or MPXV-infected individuals.

## Results

### Mpox convalescent individuals and smallpox vaccinees mount a highly shared but distinct serological reactivity towards poxvirus antigens

In total 27 poxvirus antigens (24 MPXV and 3 VACV antigens), spanning diverse functions from structural viral proteins to those involved in virion morphogenesis and host immunomodulation (Table 1) were tested against a panel of serum samples of those with prior Smallpox-vaccination or MPXV infection (Fig. 1, Table 1).

Of the negative samples, individuals demonstrated minimal antibody binding to all the MPXV or VACV antigens, with the exception of one individual, showing binding to the MPXV H3 and VACV A27 antigen. Binding to MPXV C18 was observed across all negative samples. In individuals that received one dose of the IMVANEX vaccine, antibody binding to MPXV and VACV antigens was generally low (post-dose 1, 21 days post-vaccination, Fig. 1), however, all individuals generated antibodies post-vaccination that were able to bind the MPXV B2 (VACV

## Table 1 | Overview of samples used in this study

| | Sample group | n = | Median time since vax/inf (days) | Range since vax/inf (days) | Median age |
|---|---|---|---|---|---|
| Negatives | IMVANEX pre-vaccination | 18 | - | - | 27 |
| | ACAM2000 pre-vaccination | 5 | - | - | N/A |
| | Paediatric negatives | 256 | - | - | 3 |
| Smallpox vaccinated | IMVANEX post-dose 1 (time point 1) | 8 | 24 | 14 | 26 |
| | IMVANEX post-dose 2 (time point 1) | 10 | 14 | 14–28 | 27 |
| | IMVANEX post-dose 2 (time point 2) | 7 | 43 | 37–63 | 26 |
| | IMVANEX post-dose 2 (time point 3) | 8 | 63 | 63–85 | 26 |
| | IMVANEX post-dose 2 (time point 4) | 7 | 84 | 84–106 | 26 |
| | ACAM2000-vaccinated | 5 | 40 | 30–200 | 23 |
| Mpox | Convalescent Mpox (Clade IIa) | 3 | 7 | 5–20 | 37 |
| | Convalescent Mpox (Clade IIb) | 43 | 81 | 20–113 | 37 |
| Confounders | Confounder: CMV (aged ≤50) | 62 | N/A | N/A | 34 |
| | Confounder: CMV (aged ≥51) | 37 | | | 63 |
| | Confounder: EBV (aged ≤50) | 54 | | | 25 |
| | Confounder: EBV (aged ≥51) | 46 | | | 62 |
| | Confounder: Rheumatoid (aged ≤50) | 82 | | | 15 |
| | Confounder: Rheumatoid (aged ≥51) | 18 | | | 63 |
| | Confounder: VZV (aged ≤50) | 58 | | | 32 |
| | Confounder: VZV (aged ≥51) | 42 | | | 60 |

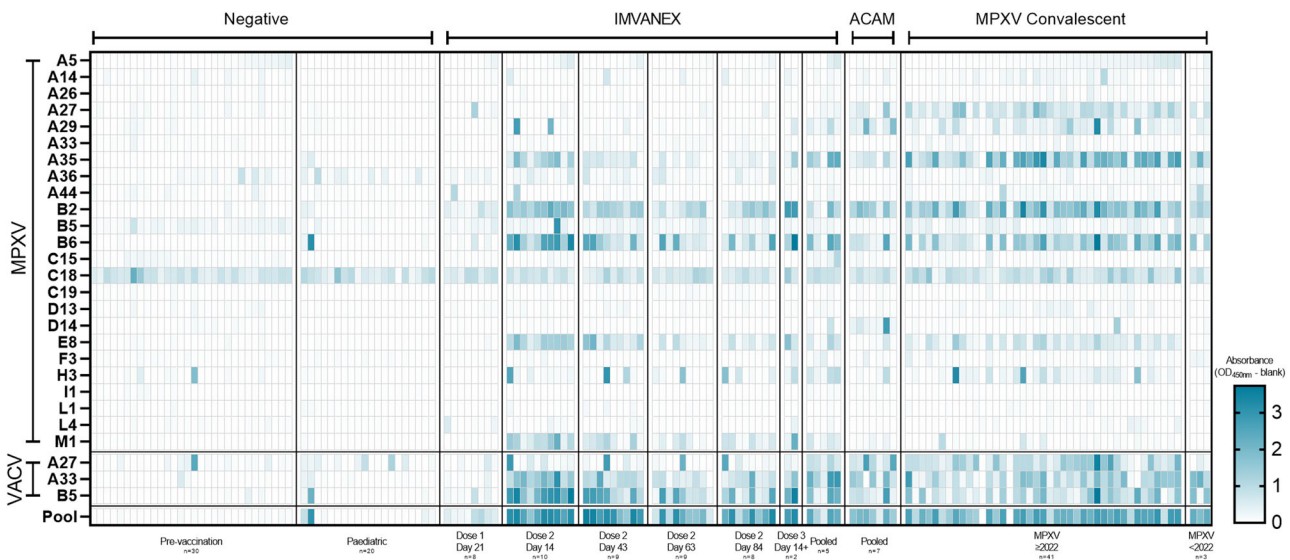

**Fig. 1 | Heatmap of ELISA results of serum samples from negative (pre-vaccination or paediatric), vaccinated (IMVANEX or ACAM2000) or MPXV-infected individuals using individual MPXV, VACV or a pool of MPXV and VACV recombinant antigens.** Colour scale represents the OD. Top panel: MPX antigens, split according to the different sample groups. Middle panel: VACV antigens, split according to the different sample groups. Bottom panel: Using a pool of four MPXV and one VACV recombinant antigens. *n* is equal to the number of biologically independent samples.

A56) antigen. Antibody binding to the VACV B5 and the MPXV homologue; B6, was variable across individuals, with only one individual demonstrated robust antibody binding to both the VACV B5 and MPXV-B6 antigens. Two-dose IMVANEX-vaccinated individuals (14 days post-vaccination) demonstrated diverse recognition of a number of MPXV and VACV antigens, with the strongest binding of antibodies observed to VACV B5 and MPXV B6 (homologous proteins), followed by MPXV antigens A35, B2, E8 and M1 (Post-dose 2, 14 days post-vaccination, Fig. 1). Some individuals had antibodies able to bind MPXV A5, A29 and H3, but absorbances were generally low, with the exception of MPXV A29 in which two individuals demonstrated strong binding. Following further time points, post-dose 2 vaccination (43-, 63- and 84 days post-dose 2), lower binding to antigens was observed, in particular to MPXV antigens B2, A35, B6, E8 and M1, and VACV A33 and B5 demonstrating similar decreasing binding.

We were also able to sample two individuals that received two historical IMVANEX doses (>3 years prior) with a booster (third) dose of IMVANEX (Post-dose 3, Fig. 1). These individuals show similar antibody binding to antigens as those in the post-dose 2 cohort, with strong antibody binding to the MPXV B6, E8, A35, H3 and M1 antigens, and the VACV A33 and B5 antigens. Binding to the M1 antigen was only observed in those receiving two or three doses of the IMVANEX vaccination. ACAM2000 Smallpox vaccination similarly induced antibodies that could bind the same antigens as IMVANEX-vaccinated individuals, with the exception of the MPXV M1 antigen, whilst binding was observed to the MPXV A27 antigen, which was not observed in IMVANEX-vaccinated individuals.

Serum samples from individuals with prior MPXV infection were similar to those from post-vaccination individuals, primarily mounting antibodies that bind VACV B5, A27 and A33 and MPXV-B2, B6, A27, A35 and E8, with variable binding across different individuals to these antigens and others such as A5, A14, A29, M1 and H3 (Fig. 1, MPXV). Serum samples from individuals with confirmed Mpox disease prior to the 2022 outbreak (Clade IIa) displayed similar antibody binding to antigens as serum samples from individuals during the 2022-2023 Mpox outbreak (Clade IIb), notably A35, A27, B2 and B6, and VACV B5, but variable binding to MPXV-A29 and A44. Notably, no difference in antigen recognition was observed between those aged <51 and >51, with the latter more likely to have had historical smallpox vaccination (Supplementary Fig. 1). Two individuals with prior MPXV infection

were found to have no antibody responses to any of the antigens tested here (Fig. 1).

Using Pearson correlation, trends in antibody binding to diverse MPXV and VACV antigens were determined (Fig. 2). Most negative samples correlated strongly with one another but some samples from individuals demonstrated poor correlation to other negative samples. However, generally, there was minimal correlation between the negative samples to other groups such as vaccinated or previously infected, with the exception of dose-one IMVANEX-vaccinated individuals, who demonstrated variable correlation in antibody binding to those post ACAM2000 vaccination, post-dose 2 IMVANEX vaccination or post-MPXV infection.

Conversely, serum samples from those post-two-dose IMVANEX vaccination at four different time points (14-, 43-, 63- and 84 days) had the strongest positive correlation observed, both between individuals and over different time points, with decreasing correlation over time since second-vaccine dose (Fig. 2). Serum from post-two-dose IMVANEX vaccinated also strongly and positively correlated with ACAM2000-vaccinated individuals and convalescent individuals with Clade IIa and IIb MPXV infection, again with similar strong correlation to one another. MPXV-infected individuals were highly correlated to one another, but as described earlier, also strongly correlated with those post-Smallpox vaccinations.

Similarly, Pearson correlation was used again to determine the correlation between MPXV/VACV antigens across the different groups (Supplementary Fig. 2). MPXV and VACV protein homologues were strongly correlated with one another in all groups as expected, including negative samples: VACV B5 with MPXV B6, VACV A33 with MPXV A35 and VACV A27 with MPXV A29, with the exception of MPXV A29 and VACV A27 in the negative samples.

Within the Smallpox-vaccinated group (IMVANEX or ACAM2000), a positive correlation was observed between a number of MPXV antigens including B2 with A35, B6, E8 and M1, as well as VACV antigens A33 and B5, and similarly observed in the mpox convalescent group, but also with MPXV A27 (Supplementary Fig. 2).

Using antibody binding data to the 24 MPXV and 3 VACV antigens, principal component analysis (PCA) was performed to determine clusters in differential binding repertoires between pre-vaccination/negative individuals and the Smallpox vaccinated (Dose 1 IMVANEX, Dose 2 IMAVENEX and ACAM2000), or with prior MPXV infection groups

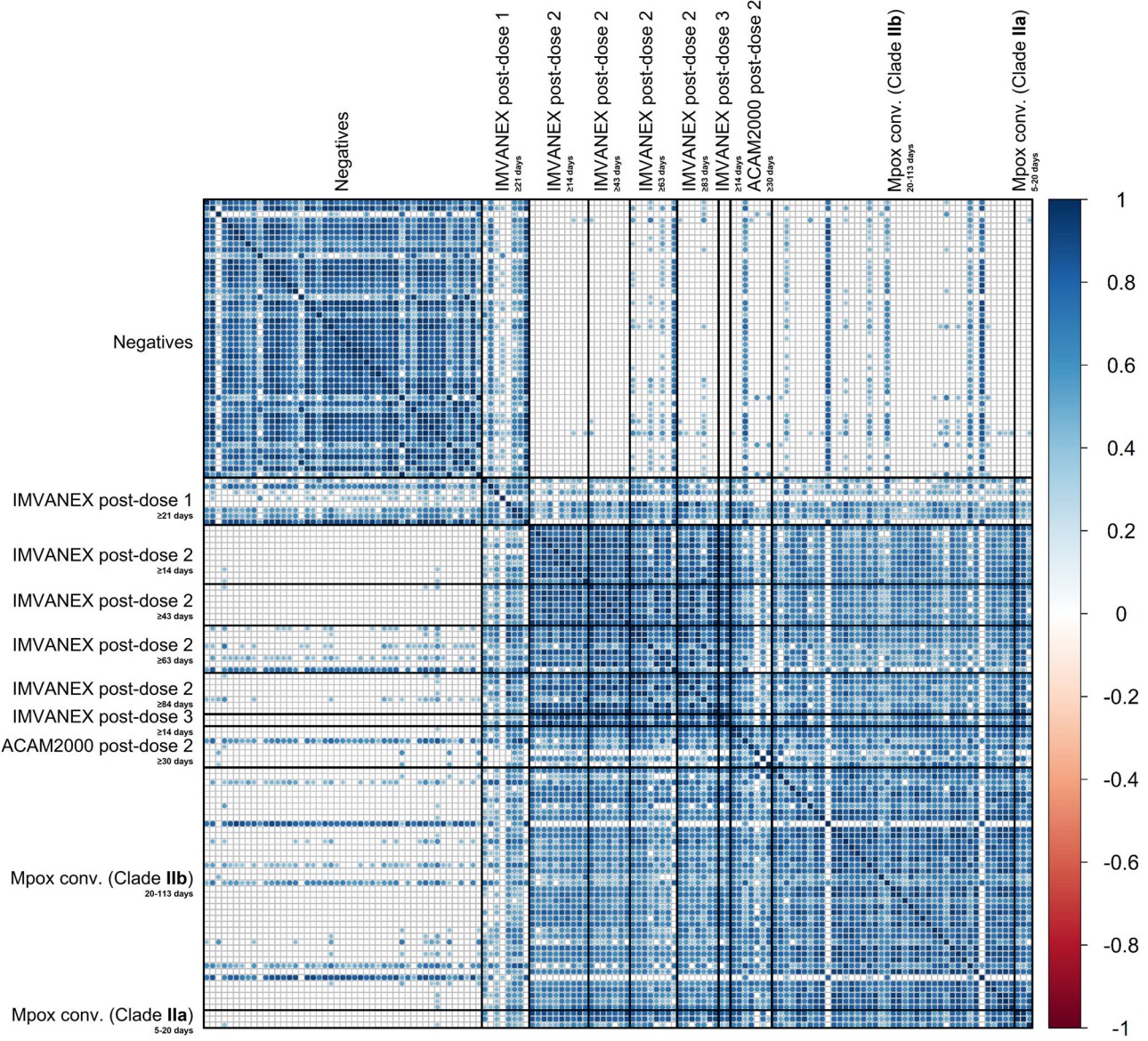

**Fig. 2 | Pearson correlation matrix of VACV and MPXV antigens in those vaccinated or MPXV-infected.** Groups include those with no prior infection or smallpox vaccination (negatives), IMVANEX vaccinated (dose 1, dose 2 and dose 3), ACAM2000-vaccinated and convalescent MPXV-infected individuals (Clade IIb and IIa). Two-tailed correlation was performed using all ELISA data for MPXV and VACV antigens. Only significant correlations are shown, with blank cells indicating a non-significant ($p \geq 0.05$) correlation. Data produced using the CorrPlot package.

(Clades IIa and Clade IIb). Antibody binding within the negative group all clustered similarly close to one another, whilst dose 1 IMVANEX vaccinated individuals similarly clustered both with one another but also with the negative samples (Fig. 3a). Dose 2 IMVANEX vaccinated individuals showed distinct clusters, different from the negative samples, however, serum samples from individuals with further time points post-IMAVNEX vaccination clustered towards the negative samples (Supplementary Fig. 3). Mpox convalescent individuals formed a distinct grouping separately from IMVANEX vaccinated or negative samples, however, displayed a degree of overlap with both of these groups. ACAM2000-vaccinated individuals were not distinctive, falling within the middle of the mpox-convalescent samples but separate (overlapping) from the IMVANEX vaccinated samples (Fig. 3a).

Using a biplot from the principal component analysis to determine individual variables driving the differentiation, individual antigens were highlighted as distinctive to particular groups (Fig. 3b). For MPXV-infected individuals, MPXV antigen A27 was the most specific to the mpox-infected and ACAM2000-vaccinated

group, followed by MPXV A14, D13 and A26, as well as VACV A27. Antigens specific to the IMVANEX-vaccinated group were primarily MPXV M1, but also included MPXV antigens E8, A36 and VACV antigens B5 and A33.

## Serological reactivity to specific MPXV and VACV antigens can discriminate between mpox convalescent from Smallpox-vaccinated individuals

Based on a number of analyses (Fig. 1, Fig. 3a, b), several antigens show promise in being used to differentiate between Smallpox-vaccinated and MPXV-infected individuals. We performed ROC analysis on some of the singular antigens identified in analyses described here (MPXV A27 and MPXV M1) to determine their feasibility in discriminating between vaccinated and infected (Fig. 4a, b, Table 3). No significant difference ($p = 0.8708$) was observed in antibody binding between the negative and IMVANEX vaccinated group for antibodies binding MPXV A27; however, MPXV-infected individuals had significantly higher antibody

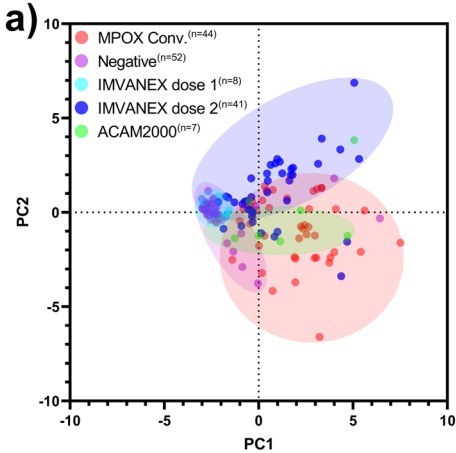

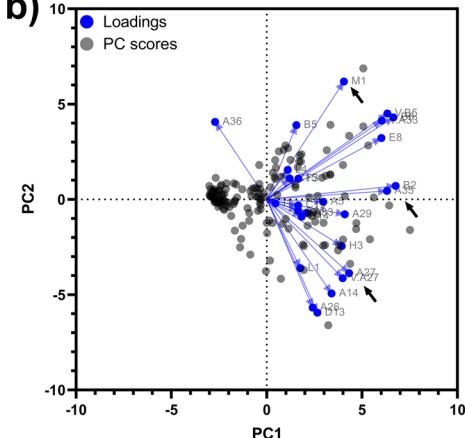

**Fig. 3 | Principal component analysis to identify similar antibody responses between IMVANEX-vaccinated and MPXV-infected individuals. a** Principal component analysis (PCA) of antibody binding to 23 MPXV and 3 VACV recombinant antigens, plotted and coloured by group. Coloured circles represent all samples within that cohort. **b** Biplot of PCA, highlighting that a number of ELISAs using specific antigens can be used for each particular group. Arrows highlighting recombinant MPXV antigens B2, M1 and A27. Note: C18L was excluded from the principal component analysis due to non-specific antibody binding. *n* is equal to the number of biologically independent samples.

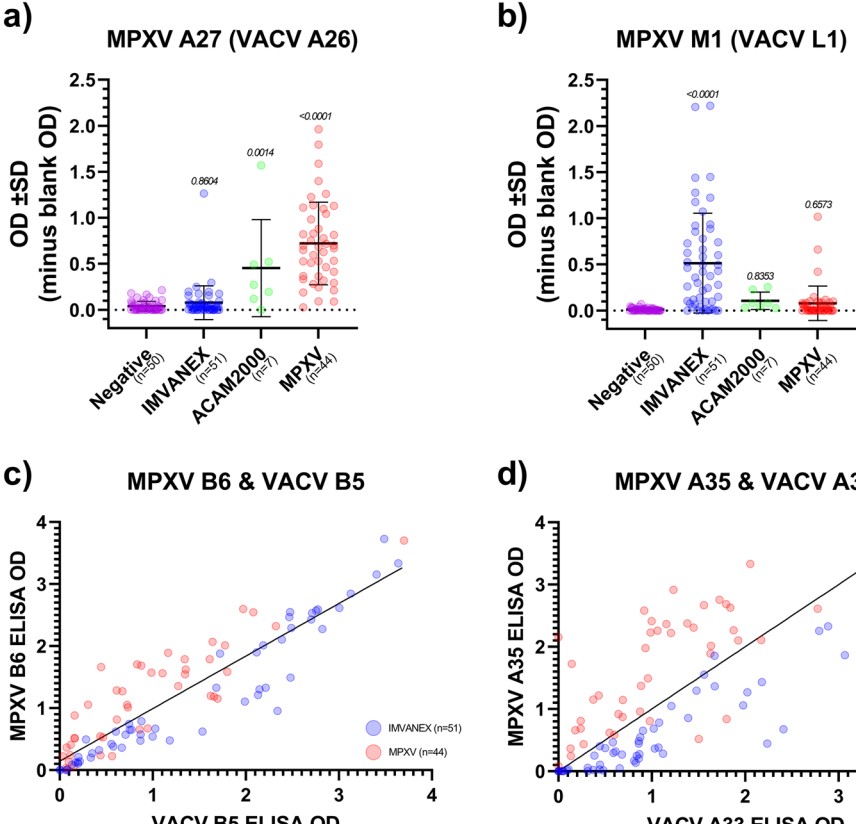

**Fig. 4 | Differential antibody responses between vaccinated and MPXV-infected individuals.** Differential MPXV poxvirus antigens used to determine: **a** Prior MPXV infection or ACAM2000 vaccination using MPXV A27 and **b** recent IMVANEX vaccination using MPXV M1. Cutoffs used are those defined in Table 3. *P* values above each column were generated using an ordinary one-way ANOVA with a multiple comparison follow-up test comparing the mean of each column with the mean of the negative column. Bars represent the mean ± standard deviation. Differential binding to the **c** VACV B5 and MPXV B6 and **d** VACV A33 and MPXV A35 protein homologues between the MPXV-infected individuals and MVA-vaccinated (IMVANEX and ACAM2000: VACV) individuals. Those with MVA vaccination show stronger binding to the VACV B5 and A33 protein compared to the MPXV B6 and A35 homologue, whilst MPXV-infected individuals show higher binding to the B6 and A35 protein. *n* is equal to the number of biologically independent samples.

responses ($p < 0.0001$) compared to negatives. Similarly, ACAM2000 individuals had significantly ($p < 0.0001$) higher antibodies to MPXV A27 than negatives (Fig. 4a). Equally, when utilising the MPXV M1 antigen, we observed that only IMVANEX-vaccinated individuals had significantly higher ($p < 0.0001$) antibodies compared with the negative samples, ACAM2000-vaccinated, or MPXV-infected individuals (Fig. 4b).

In our initial experiments, we observed that Smallpox-vaccinated and MPX-infected individuals had similar binding to both the VACV B5 and the MPXV homologue B6 (Table 2). Separating the Smallpox-vaccinated and the MPXV-infected groups, we observed preferential binding to the protein that individuals were either vaccinated or infected with, whereby Smallpox-vaccinated individuals bind VACV B5 better than the MPXV homologue B6 and skewed towards the B5 (VACV) axis (Fig. 4c). Conversely, MPXV-infected individuals were found to bind MPXV B6 better than the VACV homologue B5, with the majority of samples skewed towards the B6 (MPXV) axis. The same was also observed for the VACV A33 and MPXV-A35 proteins (Fig. 4d); however, there was minimal antibody binding to either poxvirus antigen observed in MVA-vaccinated individuals and only in MPXV-infected individuals. These data suggest antigen exposure influences the subsequent antibody binding to MPXV or VACV antigens.

**Only a subset of MPXV and VACV antigens are recognised longitudinally following IMVANEX vaccination, with waning observed**

Using a time course from IMVANEX vaccinated individuals, we assessed the antibody binding in these individuals; prior to vaccination, 28-days post-primary vaccination dose (post-dose 1 (PD1) D24), and 14, 43, 63, 84-, 122-, 157- and 185-days post-dose 2 (PD2). The majority of individuals demonstrated no antibody binding to the majority of antigens tested, demonstrated in earlier observations (Fig. 1, Fig. 5), however, antibodies were able to bind nine antigens (MPXV antigens B6, B2, E8, A35, M1, A29 and VACV antigens: B5 and A33) across the time course (Fig. 5). Antibodies induced after one dose of vaccination were minimal, primarily to MPXV B2, however, induction of antibodies to these nine antigens were observed 14 days post-dose 2. Cross-reactive antibodies to MPXV C18 were observed across all time points.

Area under the curve (AUC) analysis identified nine antigens with a high AUC for use in detecting antibodies to both MPXV infection and Smallpox vaccination, with the exception of A27, which was likely MPXV-specific (Table 3). Using Dunnett's multiple comparison tests, we also identify a number of antigens that are significantly higher at the different time points post-IMVANEX vaccination or MPXV infection compared to negative samples (Table 3). In particular, a number of MPXV antigens were significantly higher 14 days post-dose 2 than

**Table 2 | MPXV antigens and their homologues in VACV and known or predicted functions**

| MPXV Protein[1] | VACV Protein[2] | VACV vs MPXV homology (%) | Type | Protein description/function |
|---|---|---|---|---|
| A5 | A4 | 95.0 | Late | 39-kDa immunodominant virion core protein |
| A14 | A13 | 92.8 | Late | IMV inner and outer membrane protein |
| A26 | - | N/A | Late | Similar to cowpox A-type inclusion protein |
| A27 | A26 (putative) | N/A | Late | N-terminal of A-type inclusion body protein of CPV, missing from MVA Bavarian Nordic (IMVANEX) strain |
| A29 | **A27** | 93.6 | Late | Intracellular mature virus (IMV) surface membrane 14-kDa fusion protein, binds cell surface heparan |
| A33 | A31 | 86.6 | - | Unknown function, 53 bp insertion in MPXV C-terminus |
| A35 | **A33** | 96.1 | Late | Extracellular enveloped virus (EEV) envelope glycoprotein, needed for formation of actin-containing microvilli and cell-to-cell spread |
| A36 | A34 | 95.2 | Late | EEV envelope glycoprotein, lectin-like, required for infectivity of EEV, formation of actin-containing microvilli and cell-to-cell spread |
| A44 | - | N/A | - | Unknown function, gene not predicted to be present in VACV Copenhagen, ACAM2000 or IMVANEX strains |
| B2 | A56 | 93.0 | Early-Late | EEV Type I membrane glycoprotein hemagglutinin, prevents cell fusion |
| B5 | B4 | 93.1 | - | Ankyrin-like |
| B6 | **B5** | 95.9 | Early-Late | Palmitated 42-kDa EEV glycoprotein required for efficient cell spread and found on membrane of infected cells and EEV envelope, complement control protein-like |
| C15 | F9 | 99.1 | Late | MV membrane protein—required for virus entry |
| C18 | F12 | 97.2 | Early-Late | IEV, actin tail formation |
| C19 | F13 | 98.7 | Late | Major envelope antigen of EEV, wrapping of IMV to form IEV, phospholipase D-like |
| D13 | C4 | 94.3 | Early | Intracellular protein—inhibits NF-κB activation |
| D14 | C3 | 93.1 | Early | Secreted complement binding protein |
| E8 | D8 | 94.7 | Late | IMV surface membrane 32 kDa protein, binds cell surface chondroitin sulfate, IMV adsorption to cell surface |
| F3 | E3 | 86.3 | Early | IFN resistance, dsRNA binding, inhibits dsRNA dependent protein kinase, and 2-5A-synthetase |
| H3 | H3 | 93.2 | Late | IMV heparan-binding surface membrane protein |
| I1 | I1 | 99.4 | Late | Virosomal protein essential for virus multiplication |
| L1 | J1 | 96.7 | Late | IMV membrane protein—participates in virion morphogenesis |
| L4 | J4 | 100 | Early | RNA pol 22-kDa subunit |
| M1 | L1 | 98.8 | Late | Myristylated IMV surface membrane protein |

[1]MPXV proteins labelled according to the genome of MPXV virus Zaire-96-I-16 (NC_003310). [2]VACV proteins labelled according to VACV Copenhagen (M35027). Data obtained from ref. 63–67. Antigens in bold are recombinant antigens used in this study.

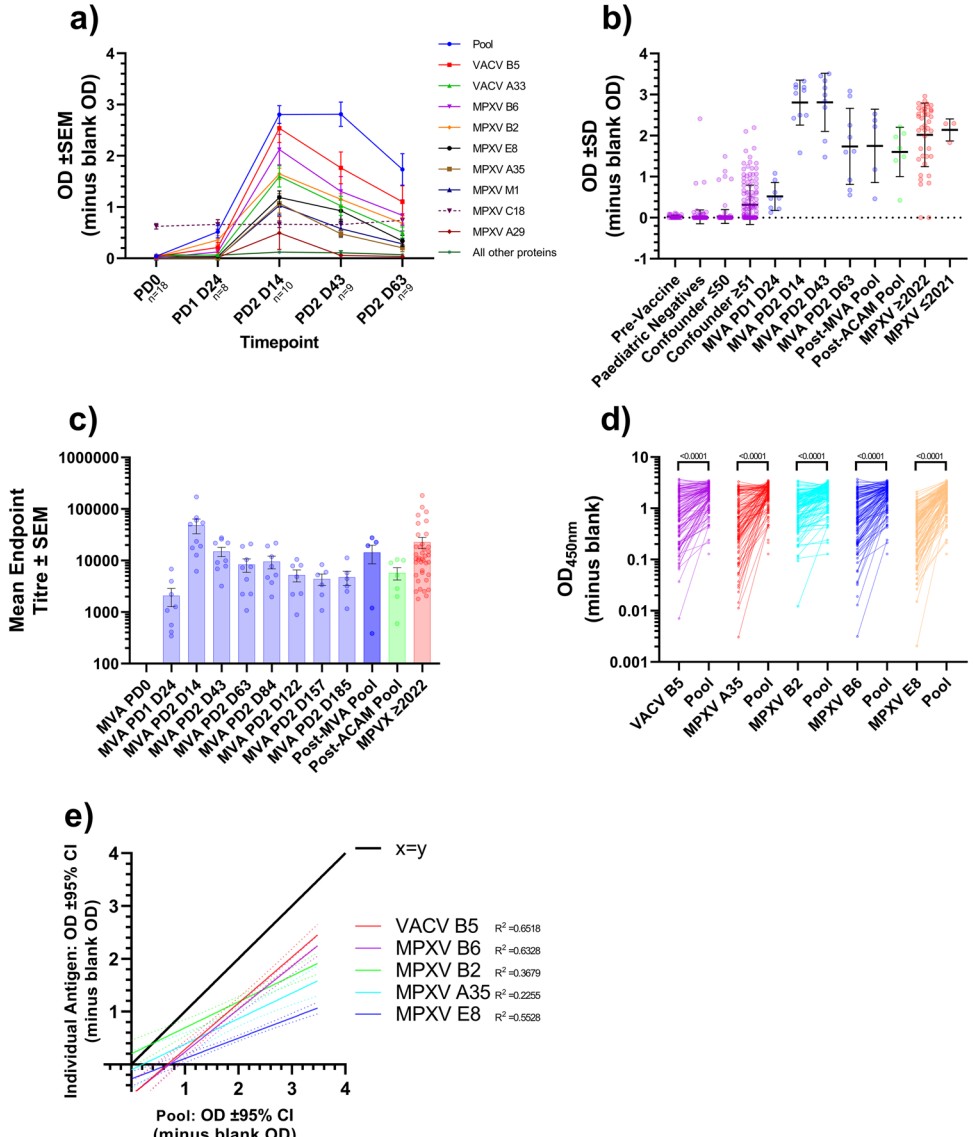

**Fig. 5 | Antibody responses to different MPXV antigens longitudinally in IMVANEX-vaccinated individuals.** Longitudinal sampling of individuals receiving IMAVENEX vaccination against: **a** a panel of 24 MPXV and 3 VACV antigens or the pooled antigen ELISA. Bars represent the mean ± standard error of the mean. *n* is equal to the number of biologically independent samples. **b** Evaluation of the pooled antigen ELISA to determine sensitivity and specificity. Bars represent the mean ± standard deviation. Each dot represents a biologically independent sample. **c** Using the pooled antigen ELISA, endpoint titres were determined for positive negative samples. Bars represent the mean ± standard error of the mean. Each dot represents a biologically independent sample. Comparison of the ELISA results of MPXV-infected and Smallpox-vaccinated serum samples using individual antigens relative to the pool of antigens (**d**) split by antigen, or **e** linear regression of individual antigen results relative to pooled antigen ELISA results (*n* = 121 biologically independent samples). Dotted lines represent the 95% confidence intervals. Black lines represent X = Y. *P* values for **d** were generated using a paired *t* test and each pair of dots joined by a line represents a biologically independent sample.

negative samples (A29 ($p = 0.0072$), A35 ($p = <0.0001$) and B5 ($p = 0.0023$) that showed no significant difference at day 63 post-dose 2. Similarly, MPXV A27 and VACV A27 were not significantly different between negatives and post-vaccination, however, they were significantly higher in mpox-convalescent individuals compared with negatives ($p < 0.0001$).

Using individual antigens for detecting post-vaccination or post-MPXV-infection was assessed using ROC analysis (Table 3). We observed that individual antigens such as MPXV B2, which had a sensitivity of 91.67% (95% CI: 78.17–97.13%) and specificity of 87.30% (95% CI: 76.89–93.42%) using a cutoff $OD_{450\,nm}$ of >0.1827, were well suited in detecting post-vaccination samples (both dose 1 and all time points for dose 2).

MPXV A27 was highly accurate in being able to discriminate MPXV-infected individuals, with ROC analysis demonstrating 91.3%

sensitivity (95% CI: 79.86% to 96.57) and 98.86% specificity (95% CI: 96.70% to 99.69%) using a cutoff $OD_{450\,nm}$ > 0.1838 (Table 3). MPXV M1 gave a sensitivity of 63.89% (95% CI: 47.58% to 77.52%) and specificity of 98.401% (95% CI 95.42% to 99.15%) in detecting IMVANEX-vaccinated individuals using a cutoff of $OD_{450\,nm}$ > 0.1164.

**Antibodies from MPXV-infected and Smallpox-vaccinated individuals can reliably be detected using a pooled antigen ELISA**
From the results of individual antigen testing, the dominant antigens recognised by MPXV-infected and Smallpox-vaccinated individuals (IMVANEX or ACAM2000) were highly similar: VACV antigen B5 and MPXV antigens A35, B2, B6 and E8. We sought to determine the feasibility of using a pool of these antigens as the basis of detecting pan-Poxvirus antibodies, preventing the requirement for the use of whole-MVA/VACV or individual recombinant pox antigens. We explored the

## Table 3 | Statistical analysis of individual MPXV/VACV antigens in detecting Smallpox- or MPXV-specific antibody responses

| Virus | Protein | Area under the ROC curve (both) | | | | Dunnett's multiple comparisons test (versus negative samples) | | | | | ROC analysis (versus negative) | | | | | | | | | | | |
|---|---|---|---|---|---|---|---|---|---|---|---|---|---|---|---|---|---|---|---|---|---|---|
| | | | | | | | | | | | Smallpox vaccine | | | | Mpox convalescent | | | | Both | | | |
| | | Total area | Std. error | 95% confidence interval | P value | PD1 D24 | PD2 D14 | PD2 D43 | PD2 D63 | MPX ≥2022 | Cutoff | Sensitivity (%) | Specificity (%) | LR | Cutoff | Sensitivity (%) | Specificity (%) | LR | Cutoff | Sensitivity (%) | Specificity (%) | LR |
| MPXV | A5 | 0.7064 | 0.04359 | 0.6210 to 0.7919 | <0.0001 | 0.998 | 0.2562 | 0.9744 | 0.9985 | <0.0001 | >0.3038 | 8.333 | 98.41 | 5.25 | >0.1903 | 34.78 | 95.24 | 7.304 | >0.1903 | 24.39 | 95.24 | 5.122 |
| | A14 | 0.6639 | 0.04553 | 0.5746 to 0.7531 | 0.0007 | 0.995 | 0.9392 | 0.4461 | 0.9814 | 0.0041 | >0.2849 | 8.333 | 98.41 | 5.25 | >0.1828 | 23.91 | 95.24 | 5.022 | >0.2661 | 9.756 | 98.41 | 6.146 |
| | A26 | 0.5494 | 0.04802 | 0.4552 to 0.6435 | 0.3091 | 0.9283 | 0.9987 | 0.9977 | 0.9918 | 0.0413 | >0.1448 | 100 | 4.762 | 1.05 | >0.1233 | 17.39 | 95.24 | 3.652 | >0.1233 | 10.98 | 95.24 | 2.305 |
| | A27 | 0.9909 | 0.005213 | 0.9807 to 1.000 | <0.0001 | 0.0394 | 0.9989 | 0.994 | 0.9981 | <0.0001 | >0.1701 | 11.11 | 98.48 | 7.306 | >0.1838 | 91.3 | 98.86 | 80.04 | >0.1383 | 58.54 | 96.58 | 17.11 |
| | A29 | 0.7243 | 0.0414 | 0.6431 to 0.8054 | <0.0001 | 0.9998 | 0.0072 | 0.9999 | >0.9999 | <0.0001 | >0.2443 | 8.333 | 98.41 | 5.25 | >0.1344 | 71.74 | 93.65 | 11.3 | >0.1475 | 42.68 | 95.24 | 8.963 |
| | A33 | 0.6155 | 0.04695 | 0.5235 to 0.7075 | 0.0173 | 0.9743 | 0.677 | 0.4456 | 0.0833 | 0.0037 | >0.1127 | 8.333 | 96.83 | 2.625 | >0.1135 | 23.91 | 96.83 | 7.533 | >0.1266 | 12.2 | 98.41 | 7.683 |
| | A35 | 0.8878 | 0.02824 | 0.8325 to 0.9432 | <0.0001 | 0.9996 | <0.0001 | 0.1738 | 0.9561 | <0.0001 | >0.2443 | 61.11 | 93.65 | 9.625 | >0.2423 | 93.48 | 93.65 | 14.72 | >0.2423 | 79.27 | 93.65 | 12.48 |
| | A36 | 0.5505 | 0.04877 | 0.4549 to 0.6461 | 0.2978 | >0.9999 | 0.5128 | 0.7373 | 0.9058 | 0.5869 | >0.1365 | 50 | 76.19 | 2.1 | <0.3010 | 97.83 | 15.87 | 1.163 | >0.1055 | 46.34 | 71.43 | 1.622 |
| | A44 | 0.567 | 0.04769 | 0.4735 to 0.6605 | 0.1676 | 0.6103 | 0.6583 | >0.9999 | 0.9997 | 0.13 | >0.1384 | 94.44 | 9.524 | 1.044 | >0.1254 | 28.26 | 90.48 | 2.967 | >0.1232 | 19.51 | 90.48 | 2.049 |
| | B2 | 0.9743 | 0.0121 | 0.9505 to 0.9980 | <0.0001 | 0.4434 | <0.0001 | <0.0001 | 0.0029 | <0.0001 | >0.1827 | 91.67 | 87.3 | 7.219 | >0.2980 | 93.48 | 93.65 | 14.72 | >0.3437 | 89.02 | 95.24 | 18.7 |
| | B5 | 0.651 | 0.04596 | 0.5609 to 0.7411 | 0.0019 | 0.9944 | 0.0023 | 0.9386 | 0.9969 | 0.7119 | <0.1165 | 75 | 58.73 | 1.817 | >0.1610 | 56.52 | 69.84 | 1.874 | >0.1275 | 64.63 | 61.9 | 1.697 |
| | B6 | 0.9383 | 0.02286 | 0.8934 to 0.9831 | <0.0001 | 0.9997 | <0.0001 | <0.0001 | 0.0078 | <0.0001 | >0.1830 | 77.78 | 95.24 | 16.33 | >0.1932 | 89.13 | 95.24 | 18.72 | >0.1830 | 84.15 | 95.24 | 17.67 |
| | C15 | 0.5407 | 0.04873 | 0.4452 to 0.6362 | 0.4045 | 0.9955 | 0.9997 | 0.9222 | 0.2299 | 0.9961 | <0.1354 | 97.22 | 11.11 | 1.094 | <0.1382 | 95.45 | 11.11 | 1.074 | <0.1382 | 96.25 | 11.11 | 1.083 |
| | C18* | 0.6045 | 0.04828 | 0.5098 to 0.6991 | 0.0323 | 0.9999 | 0.9999 | >0.9999 | 0.9612 | 0.2184 | >0.3874 | 88.89 | 31.75 | 1.302 | >0.3943 | 88.64 | 31.75 | 1.299 | >0.3874 | 88.75 | 31.75 | 1.3 |
| | C19 | 0.5853 | 0.04734 | 0.4925 to 0.6781 | 0.0789 | 0.8182 | 0.9722 | 0.8888 | 0.0017 | 0.3028 | >0.1193 | 2.778 | 98.41 | 1.75 | >0.1193 | 6.522 | 98.41 | 4.109 | >0.1193 | 4.878 | 98.41 | 3.073 |
| | D13 | 0.6453 | 0.04581 | 0.5555 to 0.7351 | 0.0028 | 0.9998 | 0.9982 | 0.5232 | 0.9976 | 0.0006 | >0.1205 | 13.89 | 92.06 | 1.75 | >0.1163 | 34.78 | 92.06 | 4.383 | >0.1163 | 25.61 | 92.06 | 3.227 |
| | D14 | 0.5681 | 0.04849 | 0.4731 to 0.6632 | 0.1603 | 0.9997 | 0.9999 | 0.9999 | 0.9997 | 0.4863 | >0.1295 | 0 | 98.41 | 0 | >0.1035 | 8.696 | 96.83 | 2.739 | >0.1035 | 4.878 | 96.83 | 1.537 |
| | E8 | 0.9123 | 0.02532 | 0.8627 to 0.9619 | <0.0001 | 0.9999 | <0.0001 | <0.0001 | 0.0118 | <0.0001 | >0.1406 | 72.22 | 98.41 | 45.5 | >0.1357 | 80.43 | 98.41 | 50.67 | >0.1357 | 76.83 | 98.41 | 48.4 |
| | F3 | 0.6087 | 0.0476 | 0.5154 to 0.7020 | 0.0259 | >0.9999 | 0.6754 | 0.9713 | 0.968 | 0.0003 | >0.1067 | 5.556 | 98.41 | 3.5 | >0.1071 | 22.73 | 98.41 | 14.32 | >0.1067 | 15 | 98.41 | 9.45 |
| | H3 | 0.6998 | 0.04355 | 0.6144 to 0.7851 | <0.0001 | 0.9983 | 0.3433 | 0.1102 | 0.8603 | 0.0185 | >0.1125 | 30.56 | 88.89 | 2.75 | >0.1349 | 60.87 | 92.06 | 7.67 | >0.1125 | 48.78 | 88.89 | 4.39 |
| | I1 | 0.6066 | 0.0514 | 0.5059 to 0.7074 | 0.0395 | 0.4004 | 0.302 | 0.4065 | 0.4221 | 0.9858 | <0.1253 | 100 | 1.961 | 1.02 | >0.1099 | 100 | 3.922 | 1.041 | >0.1099 | 100 | 3.922 | 1.041 |
| | L1 | 0.5136 | 0.04843 | 0.4187 to 0.6086 | 0.7786 | 0.9364 | >0.9999 | 0.1092 | 0.9347 | 0.9707 | >0.1615 | 97.22 | 1.587 | 0.9879 | >0.1052 | 6.522 | 95.24 | 1.37 | >0.1388 | 6.098 | 98.41 | 3.841 |
| | L4 | 0.5124 | 0.04802 | 0.4183 to 0.6065 | 0.7985 | 0.0015 | 0.6977 | 0.9759 | 0.9874 | 0.5095 | >0.1263 | 13.89 | 98.41 | 8.75 | >0.2919 | 97.83 | 0 | 0.9783 | >0.1053 | 15.85 | 96.83 | 4.994 |
| | M1 | 0.8097 | 0.05218 | 0.7074 to 0.9120 | <0.0001 | 0.9997 | <0.0001 | <0.0001 | <0.0001 | 0.1207 | >0.1164 | 63.89 | 98.01 | 32.07 | >0.1154 | 11.11 | 98.01 | 5.578 | >0.1154 | 34.57 | 98.01 | 17.35 |
| VACV | A27 | 0.7754 | 0.03866 | 0.6996 to 0.8511 | <0.0001 | 0.9998 | 0.4736 | 0.6861 | 0.9869 | <0.0001 | >0.1352 | 38.89 | 72.58 | 1.418 | >0.3706 | 80.43 | 90.32 | 8.312 | >0.3706 | 53.66 | 90.32 | 5.545 |
| | A33 | 0.9185 | 0.0252 | 0.8691 to 0.9679 | <0.0001 | 0.9998 | <0.0001 | <0.0001 | 0.0449 | <0.0001 | >0.2106 | 75 | 94 | 12.5 | >0.1387 | 84.78 | 94 | 14.13 | >0.1387 | 80.49 | 94 | 13.41 |
| | B5 | 0.9408 | 0.02086 | 0.8999 to 0.9817 | <0.0001 | 0.983 | <0.0001 | <0.0001 | <0.0001 | <0.0001 | >0.1657 | 86.11 | 92.06 | 10.85 | >0.1138 | 86.96 | 90.48 | 9.13 | >0.1138 | 87.8 | 90.48 | 9.22 |
| Pooled anti-gen ELISA | | 0.9834 | 0.01037 | 0.9630 to 1.000 | <0.0001 | <0.0001 | <0.0001 | <0.0001 | <0.0001 | <0.0001 | >0.1926 | 97.22 | 98.22 | 54.77 | >0.2350 | 95.65 | 98.22 | 53.88 | >0.1926 | 96.34 | 98.22 | 54.27 |

ROC analysis was performed for each assay (single antigen and pooled antigens) comparing the absorbance values of all negative samples versus either post-Smallpox vaccine samples, mpox convalescent or both sample groups combined. The sensitivity, specificity and likelihood ratio (LR) values for each group, alone or combined, are reported for chosen absorbance cutoffs. Area under the ROC curve values are reported for both sample groups combined apart from MPXV A27 and MPXV M1R where values for mpox convalescent samples only and post-Smallpox vaccine samples only are reported, respectively. Dunnett's multiple comparisons test was used for each group, relative to the negative control samples.* - non-specific binding was observed to C18.

sensitivity and specificity of this pool and comparisons to individual antigen results.

A number of antibodies against specific antigens increased post-dose 2, with waning observed, whilst binding to some antigens was lost at further time points post-vaccination such as MPXV A29 (Fig. 5a). Compared to individual antigens, higher ODs were obtained when samples were tested using the pool antigen ELISA, compared with individual antigens, with trends in increasing antibody titres following a second dose of vaccination, and waning observed 43 days post second dose (Fig. 5a–c). This sensitivity and specificity of this pooled antigen ELISA was then further explored, to determine feasibility in detecting antibodies induced both by Smallpox vaccination and/or MPXV infection: with an overall sensitivity of 96.34% (95% CI: 89.79% to 99.00%) and specificity of 98.22% (95% CI: 96.66% to 99.06%) determined using ROC analysis (Table 3) with an OD cut off of 0.1926. This sensitivity and specificity were based on testing the pooled antigen ELISA against a total of 613 samples: 508 negatives (paediatric negatives, pre-vaccination samples and confounders) and 105 positives (IMVANEX vaccinated (Dose 1 or Dose 2), ACAM2000 vaccinated or MPXV infected (Clade IIa and IIb)) (Fig. 5b). All vaccination and MPXV-infected samples tested positive by the pooled antigen ELISA, with the exception of some post-dose one vaccinated individuals (though were marginally below the 0.1926 cut off) and two MPXV-infected individuals, however these MPXV-infected individuals demonstrated no antibody binding to any of the MPXV/VACV antigens (Fig. 4, with samples taken 20- and 76-days post-infection).

AUC analysis identified the pool antigen ELISA as the highest AUC compared to any individual antigen but was also the only ELISA that show significantly higher results in post-dose 1 vaccinated individuals compared to negative samples, further demonstrating the utility of this pooled antigen ELISA (Table 3) as a screening assay to detect pan-poxvirus antibodies. This assay could be further quantified by performing endpoint titres to quantify and compare samples between runs, serially diluting samples 1:4 until the cut-off of 0.1926 was achieved (Fig. 5c, Table 3).

When using a combined antigen pool, this resulted in a potentiation of antibodies detected by ELISA (Bottom panel, Fig. 1). In general, all samples tested on the individual antigens had lower antibodies determined by OD measurement than the pooled antigen ELISA (Fig. 5d), with linear regression models similarly demonstrating higher ODs in the pooled antigen ELISA relative to the individual antigens (Fig. 5e). When comparing ROC analysis, the highest sensitivity and specificity for detecting antibodies induced by Smallpox vaccination or MPXV-infection was in fact the pooled antigen ELISA, with 97.22% and 98.22% sensitivity, and 95.65% and 98.22% specificity, respectively (Table 3).

## Discussion

MPXV was detected in a wide range of countries during the 2022-2023 outbreak after the initial identification of cases within the UK. Vaccination strategies have been implemented through a number of public health agencies worldwide as means to limit the spread of the disease and protect individuals from infection[40]. Here, we have demonstrated that antibody responses induced by two widely used and licensed Smallpox vaccines (IMVANEX and ACAM2000, post-dose 1, 2 and 3) and prior MPXV infection (both Clade IIa or the current 2022-2023 Clade IIb outbreak) are similar and result in antibodies able to bind a number of MPXV and VACV antigens. We also describe the development of a pooled antigen ELISA to study both Smallpox vaccine and MPXV-infection antibody responses.

### Analogous antigen recognition is observed between Smallpox-vaccinated and Monkeypox-convalescent individuals

Using an array of 27 different poxvirus antigens (24 MPXV and 3 VACV), we observe that Smallpox-vaccination (be it IMVANEX or ACAM2000)

induces a similar antibody response to those previously infected with MPXV, both Clades IIa and IIb. Minimal antibodies to the diverse poxvirus antigens were observed in negative samples, with all negative samples correlating positively to one another (Fig. 2, Supplementary data). Surprisingly, however, those with one dose of IMVANEX vaccination demonstrated minimal antigen binding, such as with negative samples, except for the presence of some antibodies able to bind MPXV B2 in one-dose vaccinated individuals, likely explaining the minimal and variable correlation observed in this group to other vaccinated or MPXV-infected groups (Fig. 2). However, variations in Eukaryotic/Prokaryotic-expressed proteins also need to be further assessed to determine effects due to glycosylation. Furthermore, differences in delivery (subcutaneous vs intradermal) and dosing (full versus fractional dosage) of the IMVANEX vaccination may also play additional roles in induction of antibody and thus antigen binding, which was not studied here and warrants further investigation.

Nevertheless, the lack of robust antibody responses after one dose of IMVANEX in previously unvaccinated individuals (aged <50) shown here does not suggest a lack of protection, as recent studies have shown that vaccine efficacy against disease after one dose of IMVANEX has been suggested to be as high as 78% when comparing vaccinated with unvaccinated individuals[41], and that antibodies may be present that are able to bind a number of antigens not measured here. Similarly, whilst one vaccination dose has been shown to induce both low antigen-binding and neutralising antibodies[42], this likely suggests a role of cellular or T-cell immunity not measured in this study but performed by others[37–39].

Within our study, we demonstrate that a number of antibodies capable of binding diverse poxvirus antigens are induced by two-dose IMVANEX vaccination or MPXV prior infection (Clade IIa and Clade IIb), with high correlation between individuals in these groups and to one another (Fig. 2). Even though we only have three Clade IIa convalescent samples, we observe no difference in antigen recognition between a Clade IIa and IIb MPXV infection, which should be further explored to understand the role of mutations in Clade IIb and their possible role in transcriptional changes and hence antigen recognition after MPXV infection[25,26]. We did observe two MPXV-infected individuals that did not show any antibody binding to any of the MPXV or VACV antigens tested here, suggesting either antibody responses to other poxvirus antigens not tested here, or minimal induction of antibodies, akin to post-dose 1 individuals.

However, whilst there is a high correlation between antibody binding to diverse MPXV/VACV antigens in two-dose IMVANEX-vaccinated and MPXV-infected individuals, distinctive but overlapping grouping is observed when performing principal component analysis (Fig. 3a) which is further narrowed with post-vaccination antibody waning (Supplementary Fig. 3). Antigen binding to poxvirus antigens is also highly similar across ACAM2000-vaccinated individuals and MPXV-infected (Fig. 2), and PCA similarly identifying overlapping groups, yet binding to the A27 antigen was observed in both groups. These data suggest a core set of antigens are shared across repeated Smallpox-vaccinations (IMVANEX and ACAM2000) and MPXV infection due to similarity across viral epitopes[37], warranting further work to understand the drivers of these shared and distinct antibody responses between two highly similar vaccines.

This aligns with the observed protection afforded when MVA was used for vaccination against MPXV infection in macaques[28,29] or used in regions with high MPXV prevalence[5]. Prior studies have demonstrated a difference in neutralising titres between those infected with MPXV aged <48 and >48[42] as a basis for historical Smallpox-vaccination, however, we see no discernible difference in antibody binding to all MPXV and VACV antigens measured in this study when using a similar age cut off (Supplementary Fig 1) but similarly confirm the low level of IgG induced by single IMVANEX vaccination. When performing principal component analysis, antibody waning in two-dose IMVANEX-

vaccinated individuals results in the harmonisation of IMVANEX vaccinated with MPXV-infected groups, preventing discrimination (Fig. 3a, Supplementary Fig. 3). Further longitudinal data is needed to understand the immunology from Smallpox-vaccination and MPXV infection, and suggests that other additional assays will be required for differentiating between MPXV-infected and VACV-vaccinated individuals such as specific-antigens or avidity assays.

Previous work has been conducted to differentiate the antibody response from MPXV infection from Smallpox-vaccination, however, this has involved cross-absorption ELISAs or a Western blot method, which identifies three diagnostic bands as a differential diagnostic[43], with none of the proteins further described. Using our data, we were able to identify immunological signatures of MPXV infection, primarily those induced to the MPXV A27 protein (Fig. 4e and Tables 2 and 3) a cowpox-like type A inclusion protein that is present in the strain used in the ACAM2000 vaccination and MPXV isolates, but missing in MVA-BN (IMVANEX) and VACV Copenhagen. The absence of A27 in VACV Copenhagen and MVA-BN may provide the ability of MPXV A27 to be used as a differential assay in serosurveillance studies in countries such as the UK that do not or have not previously used the ACAM2000 smallpox vaccine. However, further work is required to ensure that cross-reactivity of antibodies to MPXV A27 is not observed in those with other *Chordopoxvirinae* infections and that an MPXV A27 assay can be used to also detect prior infection with MPXV Clade I that was not assessed here. Similarly, we also identify MPXV M1 (VACV homologue L1) as a serological marker of IMVANEX vaccination (Fig. 4b), but antibodies able to bind M1 were observed in some MPXV-infected individuals, possibly due to prior Smallpox-vaccination. The reasoning for MPXV M1 (VACV L1), an IMV surface membrane protein that has shown to be protective against poxvirus challenge[44–46], as a differential between IMVANEX vaccinated and MPXV-infected or ACAM2000 vaccinated is unknown, and warrants further investigation, as it is present in all VACV and MPXV strains, with >98% amino acid identity between VACV Copenhagen and MPXV Zaire, suggesting a possible overexpression of this gene (and thus strong immune induction to the protein) within the IMVANEX-strain.

Similarly, whilst individual antigens show distinct and differential binding, binding between VACV and MPXV homologous antigens is different depending on whether individuals having VACV vaccination or an MPXV infection (Fig. 4c, d). This is further supported by recent work that demonstrated Smallpox-vaccine naïve individuals (e.g. those aged <48) with MPXV infection neutralise MPXV better than VACV[42], reinforcing a virus-specific immunological response that should be further explored in those with heterologous antigenic exposures.

We also observe waning in vaccine-induced antibodies at day 43 post two vaccine doses as would be expected, with a decrease in antibodies able to bind the diverse antigens (Figs. 1, 2 and 5). The further extent of this antibody and antigen-specific waning is to be explored, as others have demonstrated waning to individual antigens[47], whilst persistence of IgG and neutralising antibodies up to 88 years post-vaccination have been observed[48]. Further work is needed to study the immunogenicity and persistence of antibodies to diverse poxvirus antigens after MPXV infection and/or IMVANEX vaccination, as means to provide data on antibody persistence and inform recommendations for new vaccines or future vaccination boosters in communities at-risk of mpox transmission or infection. We are now also pursuing an understanding of the immunological impact of heterologous (both full and partial dosage) antigen exposure in those with MPXV infection followed by two IMVANEX doses, to understand antibody persistence and repertoire.

The presence of antibodies able to bind MPXV C18 (VACV F12), an IEV actin tail formation protein[49], across all samples suggests either cross-reactivity of antibodies to the recombinant antigen or non-specific sequestration of immunoglobulins. Further work is ongoing to understand antibody binding and MPXV C18.

## A pooled MPXV and VACV antigen ELISA is more sensitive and specific than individual poxvirus antigens

To perform reliable immunological studies on Smallpox vaccination, MPXV infection or serosurveillance, standardised and highly congruous ELISA assays are needed. Whilst a number of poxvirus ELISAs use whole-MVA inactivated VACV, or Western blot-based methods, these methods require virus growth and quality control with batch-to-batch variability. Using the data from our antigen panels, we describe the development of a pooled-antigen ELISA using highly purified and commercially available recombinant antigens to study Smallpox-vaccine- and MPXV-induced antibody responses.

Using a number of pre-vaccination/negative samples, longitudinal samples from IMVANEX vaccinated individuals, those with prior ACAM2000 vaccination and also MPXV-infected individuals, we demonstrate the use of nine antigens in detecting antibody responses in each of these groups (Table 3, MPXV antigens B6, B2, E8, A35, M1, A29 and VACV antigens: B5 and A33). Whilst some of these antigens are highly sensitive and specific to detecting Smallpox-vaccination- or MPXV-infection-induced antibodies, the highest sensitivity and specificity achieved was a pool of five antigens (MPXV B2, B6, E8, A35 and VACV B5). This pooled antigen assay demonstrates high sensitivity and specificity but also potentiation in antibodies detected than singular antigen ELISAs (Fig. 1, Fig. 5d, e). Furthermore, quantitation through endpoint titres is possible and enables quantification of antibody responses following vaccination and MPXV infection and the monitoring of antibody waning/long-term IgG persistence (Fig. 5c).

The high sensitivity and specificity of the pool are explained by the recombinant antigen components: MPXV B2 (VACV homologue A56) is an EEV Type I membrane glycoprotein haemagglutinin that is integral for in vivo and in vitro spread of VACV, as well as binding additional VACV proteins in infected cell membranes[50–52]. MPXV E8 (VACV homologue D8) is an IMV surface membrane protein, shown to be involved in virus entry through binding chondroitin sulphate, eliciting strong humoral immunity[53,54]. MPXV B6 (and VACV homologue B5) is a 42-kDa EEV outer surface antigen, previously shown to be a major target of EEV-neutralising antibodies[44,55,56], likely explaining the immunogenicity observed to MPXV B6 and VACV B5 in this study. Similarly, MPXV A35 (VACV homologue A33) is an EEV envelope protein and also a source of EEV protective antibodies[44,55], whilst also previously demonstrated to be associated with B5[57]. Previous studies have also demonstrated anti-A33 (MPXV A35) and anti-B5 (MPXV B6) monoclonal antibodies (mAbs) as protective against lethal VACV-challenge in mice[58,59], however, combinations of anti-VACV mAbs afforded the highest protection. These proteins are highly conserved between VACV and MPXV[46] (Table 2), likely suggesting minimal changes in assay sensitivity if VACV homologues were used; however, based on differential binding demonstrated here (Fig. 4c, d), this should be further explored.

As a number of antibodies are induced to these antigens due to MPXV infection, further work is warranted to determine if a combination of these antigens provides protection from disease as next-generation Orthopoxvirus vaccines. Prior work has similarly demonstrated that some of these antigens (both individually or combined) induce a robust antibody response when used in an mRNA[60,61] or DNA[44–46] vaccine, some of which have demonstrated protection against disease. Similarly, monoclonal antibodies against some of these antigens may provide new post-exposure or long-lasting therapeutics for mpox or broad poxvirus disease, in replacement of convalescent serum such as Vaccinia immune globulin (VIG) or recombinant VIG[58,59,62], something we are actively exploring.

One of the major strengths of this study is that we used a large number of recombinant MPXV/VACV antigens, some of which are produced in mammalian cells, which are more representative of viral proteins (including glycosylation) than linear peptides used in protein arrays. Furthermore, using 24 MPXV antigens, these cover ~15% of

proteins encoded within the MPXV genome. Further work is now ongoing to perform expression and purification of all MPXV antigens to determine the range of antibodies involved in antigen binding induced by infection/vaccination using customised-multiplex assays (e.g., Luminex) to determine binding to whole proteins rather than linear epitopes using protein microarrays. Additional VACV homologues are also being sought to provide additional information on the preferential binding observed (Fig. 4).

Our pooled antigens are skewed towards MPXV, though the data demonstrated here show that this pool is the only assay (compared to individual antigens) capable of detecting a statistically significant increase in antibodies relative to negative controls. Further optimisation of this antigen pool is ongoing to determine optimum antigens for use in vaccination immunology studies and serosurveillance.

Similarly, we did not conduct MVA, VACV or MPXV neutralisation as part of this study; however, our observations of low antibodies induced post one-dose of IMAVENX correlates to observations by others[42] and is observed when using endpoint titres using the pooled antigen ELISA. Further work is needed to understand the role of binding and neutralising antibodies in the context of correlates of protection.

Finally, the samples we used within this study were a diverse mix of vaccination/mpox samples from individuals at differing time points; however further time points are required to monitor waning in both the vaccination and MPXV-infected individuals, as well as assessing antibody persistence in those with heterologous antigenic exposure.

Here, our results provide a wealth of information on the immunology of MPXV infection and also demonstrate analogous humoral antigen binding between those with Smallpox-vaccination (IMVANEX or ACAM2000) and those with prior MPXV infection (both Clade IIa and IIb). Furthermore, the singular proteins identified in this study in the immunological responses to both Smallpox vaccination and MPXV infection may offer new targets for future vaccination strategies (e.g., mRNA vaccines) or therapeutics (e.g., mAbs). Similarly, using the analogous immune responses between Smallpox-vaccinated and MPXV-convalescent, we developed a highly sensitive and specific pooled antigen ELISA that offers a mechanism for reliably measuring immune responses post-infection or post-vaccination without the need for whole-virus ELISAs or live-virus neutralisation. Taken together these findings offer an opportunity for the development and assessment of next-generation Orthopoxvirus vaccines.

# Methods

### Antigens
Recombinant MPX protein antigens were sourced from several commercial companies, as either Eukaryotic- or Prokaryotic- expressed proteins (Supplementary Table 1) as follows: A14, A26, A27, A29, A36, A44, B5, B6, C15, C18, D13, D14, E8, F3 and L1 (ProteoGenix, France), B2 and C19 (Abbexa, UK), A5L and L4R/VP8 (Native Antigen Company, UK) and A33, A35, H3, I1 and M1 (SinoBiological, China). MPXV sequences for recombinant protein expression were all based on Clade II. Additionally, recombinant VACV Copenhagen protein antigens A27, A33 and B5 were sourced from SinoBiological.

### Single antigen ELISAs
ELISAs utilising single MPXV or VACV antigens were all performed using the same method, with variation in the coating antigen only. Briefly, 48 wells of a Corning High-Binding 96-well plate were coated with 100 μl/well of 0.1 μg/ml recombinant antigen overnight at 4 °C, with the other 48 wells 'coated' with PBS (Gibco, 10010). After overnight incubation, plates were washed three times with 300 μl/well of PBS with Tween20 (0.01% final) using a Biotek 405 TS plate washer before being blocked with 200 μl SuperBlock (ThermoFisher, 37516) for 60 minutes at room temperature. Plates were washed as before, serum samples were diluted to 1:200 in Superblock and 100 μl per well

was applied to both the antigen-coated and PBS-only wells. After a 60-minute incubation with the samples at 37 °C, plates were washed as before and 100 μl/well goat anti-human IgG preabsorbed H + L (AbCam, ab98624), diluted 1:8000 in Superblock, was added to the plate. Plates were incubated for a further 60 minutes at 37 °C, followed by a wash with 300 μl/well of PBS with 0.01% Tween20 five times. Plates were then developed with 50 μl/well 1-Step™ Ultra TMB-ELISA Substrate Solution (ThermoFisher, 34029) for 30 minutes at 37 °C and stopped with 50 μl/well KPL TMB Stop Solution (Seracare, 5150-0021). Plates were then read at an absorbance of 450 nm using an Infinite F50 plate reader (Tecan, 30190077).

### Pooled antigen ELISA
A pooled-antigen ELISA was performed similarly to single antigen ELISAs described above, however, wells were coated with 100 μl/well of a pool of recombinant MPXV antigens A35, B2, B6 and E8, and recombinant VACV antigen B5, each at a concentration of 0.1 μg/ml. In addition, to determine endpoint titres, samples were also serially diluted 1:4 seven times in SuperBlock, starting at a 1:100 dilution and 100 μl added per well.

### Negative and confounder samples
To determine pox-antigen reactivity and background pox-antibody reactivity, paediatric serum samples were sourced from the UK Health Security Agency (UKHSA) Seroepidemiology Unit (SEU), Manchester ($n = 256$) to develop the assays and ensure specificity. In addition, confounder serum samples from those with PCR-confirmed infection with CMV ($n = 99$), EBV ($n = 100$) and VZV ($n = 100$) along with serum samples positive for Rheumatoid factor ($n = 100$) were used to determine assay specificity.

### Serum samples from smallpox-vaccinated individuals
Individuals with prior or imminent IMVANEX vaccination were recruited from UKHSA Porton Down. Individuals in the process of receiving IMVANEX vaccination were bled before their primary intramuscular vaccination ($n = 18$), then further bled at 24 days post dose one ($n = 8$) and 14 ($n = 10$), 43 ($n = 7$), 63 ($n = 8$) and 84 days ($n = 7$) post dose two. Two individuals with historical IMVANEX vaccination (>4 years prior) receiving a booster were bled 14 days post dose ($n = 2$). In addition, serum samples were provided by the Poxvirus and Rabies Branch at the US Centre for Disease Control and Prevention (US-CDC) from individuals post ACAM2000 single dose vaccination ($n = 4$) and multiple-dose vaccination ($n = 1$). Further detail can be found in Table 1.

### Mpox convalescent serum
Residual and anonymised Mpox convalescent serum from PCR-confirmed cases from the 2022 Clade IIb outbreak ($n = 43$) were obtained for diagnostic assay development from the Rare and Imported Pathogens Laboratory, Imperial College Healthcare NHS Trust and Chelsea and Westminster Hospital NHS Foundation Trust. Year of birth was used as a proxy for historic Smallpox vaccination (<1971), although noted that not all patients born prior to the end of the eradication programme may have been vaccinated. Days since diagnosis were also recorded. Convalescent Mpox Clade IIa sera from 2018 and 2019 ($n = 3$) were obtained from the Rare and Imported Pathogens Laboratory from previously imported Mpox cases.

### Data analysis
Data processing and handling were done using Microsoft Windows Excel (version 2208 (15602.30578)), GraphPad Prism software (version 9.2.0(33)). ELISA absorbance data was handled using Tecan Magellan Software (Version 7.5). Data analysis was achieved using the built-in analysis tools of GraphPad Prism (version 9.2.0, as above), with no modifications.

Before analysis, the absorbances achieved on the PBS-only wells were subtracted from the corresponding absorbances achieved on the antigen-coated wells. All data analysis (curve fitting, receiver operator curve (ROC), or principal component analysis was performed using GraphPad Prism 9.2.0 (GraphPad Software, USA) unless otherwise stated. For Pearson correlation CorrPlot (version 0.92) using R (Version 4.02) and R Studio (1.3.1056). Analysis was performed whereby $r$ was computed for every pair of $y$ data set and used to generate a correlation matrix. Data were assumed to have Gaussian distribution with $p$ values determined using the Two-tailed method. Principal component analysis was performed using the OD of each antigen for each sample as a continuous variable and principal components were selected based on parallel analysis with 1000 simulations performed at a 95% percentile level. The results of the principal component analysis were grouped by sample group. The endpoint titres of a sample were calculated by fitting a sigmoidal 4PL model to sample absorbances versus log[10] transformed sample dilution and interpolating the dilution at an absorbance of 0.1926.

## Ethics

Samples from ACAM2000-vaccinated individuals were collected at the Centre for Disease Control and Prevention through IRB #3349. For testing herein, samples were additionally deidentified under a separate IRB-approved protocol (#7294). Samples obtained from IMVANEX Smallpox-vaccinated individuals were obtained through written and informed consent through UKHSA Research and Ethics Committee ("REGG") for assay validation. NHS Research Ethics Committees (REC) granted approval for sampling from previous MPXV-infected individuals under reference 22/HRA/3321, with residual and fully anonymised serum from individuals with prior MPXV infection sourced from diagnostic laboratories for surveillance, assay performance assessment, validation and public health monitoring.

## Reporting summary

Further information on research design is available in the Nature Portfolio Reporting Summary linked to this article.

## Data availability

All serology data generated in this study are provided in the Supplementary Information and Source Data file. Genomes of MPXV and VACV are publicly available through NCBI under accession numbers: MPXV virus Zaire-96-I-16 (NC_003310) and VACV virus Copenhagen (M35027). Source data are provided with this paper.

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

## Acknowledgements

We would like to thank members of the CEPI, in particular Valentina Bernasconi, Mark Manak and Ali Azizi for their guidance throughout the development of the pooled antigen ELISA assay. This work was funded by the UK Health Security Agency, with additional funding obtained from the Coalition for Epidemic Preparedness Innovations (CEPI) for the development of a mpox/Smallpox ELISA assay.

## Author contributions

A.O. was involved in the conception of the work, supervision of testing, oversight of the project, manuscript writing and review. S.J., B.H. and H.S. performed the serology testing and contributed to manuscript writing and review. D.B., H.C., C.H., T.R. and N.C.G. provided samples, and contributed to the research and manuscript writing. P.S.S., M.T. and I.D. provided ACAM2000 samples, advice regarding assays and pox immunology and contributed to manuscript drafting. R.M., M.P. and R.J. provided samples. D.W. performed phlebotomy and sample gathering. C.O., S.T. and E.L. provided samples. G.H., T.C., S.M. and A.L. performed sample receipt and testing. T.B., R.V., C.R. and B.H. oversaw supervision and contributed to manuscript writing. All authors read and reviewed the manuscript before submission.

## Competing interests

The findings and conclusions in this report are those of the authors and do not necessarily represent the official position of the UK Health Security Agency or the Centres for Disease Control and Prevention. The authors declare no competing interests.

## Additional information

[1]Emerging Pathogen Serology group, UK Health Security Agency, Porton Down, Wiltshire, UK. [2]Rare and Imported Pathogens Laboratory, UK Health Security Agency, Porton Down, Wiltshire, UK. [3]Department of Infection and Immunity, University College London, London, UK. [4]The Hospital for Tropical Diseases, University College London Hospital, London, UK. [5]NIHR University College London Hospitals BRC, London, UK. [6]Poxvirus and Rabies Branch, Centre for Disease Control and Prevention, Atlanta, GA, USA. [7]Imperial College Healthcare NHS Trust, London, UK. [8]Chelsea and Westminster Hospital NHS Foundation Trust, London, UK. [9]Research and Development, UK Health Security Agency, Porton Down, Wiltshire, UK. [10]Immunisation and Vaccine Preventable Diseases Division, UK Health Security Agency, Colindale, London, UK. [11]Seroepidemiology Unit, UK Health Security Agency, Manchester, UK. [12]These authors contributed equally: Ashley D. Otter, Scott Jones, Bethany Hicks. ✉e-mail: ashley.otter@ukhsa.gov.uk

