## [Peer Review File · Nature Communications]

Reviewers' Comments:

Reviewer #1:

Remarks to the Author:

The main findings of the work by et al. is cross-reactivity between antibody responses directed at monkeypox and vaccinia/smallpox vaccination/infection(s). This finding is not new and dates back to the first descriptions of monkeypox. However, the authors do describe it with the newest strain. The authors also carry out a rather comprehensive serological analysis of monkeypox antigens (less so with vaccinia), and this is the strength of the work. They aim to try to differentiate the responses and find antigen responses or signatures that may help in this. However, this problem has been tackled before by Mark Slifka (<https://www.ncbi.nlm.nih.gov/pmc/articles/PMC2570942/>) which seemm or claims to have solved this issue. No reference or comparison is given to this work. Nonetheless, I feel the work presented in solid and adds substantially to what is known about individual responses to the various proteins, but may be better suited to a more specialized journal.

Other minor problems include lack of description of how negative cut-offs determined, as well as functional studies of the antibodies, which the authors point to as well.

Reviewer #2:

Remarks to the Author:

Otter et al. performs a comprehensive serological assessment of individuals previously infected with monkeypox or that had received varying doses of two different smallpox virus vaccines (IMVANEX or ACAM2000) and compared them to a large control/confounder cohort to assess serological reactivity (IgG ELISA) to 24 mpox and 3 vaccinia virus antigens at different time points. I believe this is extremely valuable information that will be a great addition to the field both in our understanding of humoral immunity to mpox virus and related vaccines as well as clinical diagnostic test development.

Major Comments:

While I appreciate the importance and quality of this data and the wealth of information presented in this manuscript, it needs to be rewritten to be more concise with less redundancy in data presentation/figures and clear-cut take away points. It appears that main text figures are repeating the same data in multiple forms of statistical/descriptive analyses that do not add much more value (more specifically, the Pearson correlation matrices and sections of text dedicated to broadly describing correlations). The manuscript main text figures should be limited to the most valuable ones with the most important messages.

For example, in Results, the first three sections should be combined into one, with a title similar to the following: "Mpox convalescent individuals and smallpox vaccinees mount a highly shared yet distinct serological reactivity towards 24 mpox and 3 vaccinia virus antigens." As it is currently written, the first and third section seem to be redundant to each other ("analogous" versus "shared yet distinct") and the second section seems unnecessary. I would use Figure 1 and Figure 3D-E as main text figures for this section. This section would highlight similarities and differences in a general sense, ending with PCA analysis.

Then, in Results, sections four and five should also be combined with a title similar to: "Serological reactivity to specific mpox/vaccinia antigens can discriminate mpox convalescents from smallpox vaccinees". This is indeed an important finding to highlight on its own, and is a great follow-up to the PCA analysis that identifies candidates for discrimination of patient groups, first Mpox A27 and Mpox M1 and then Mpox B6 vs.VACV B5 and Mpox A35 vs.VACV A33, which is a great finding. For this I would combine Figures 4 and 5.

Then, section six should remain on its own as a very nice longitudinal analysis, as well as section seven with the identification of an optimal antigen pool for detection of antibodies in mpox convalescents and smallpox vaccinees.

Again, this is a wonderful paper, but it was difficult to follow and required several re-reads to fully understand specific take-away points. Re-writing/organization would greatly improve the quality to

match the high-quality data being presented!

Minor Comments:

(1) Could you comment on the utility (or lack thereof) in including IgA and/or IgM antibodies or a pan IgG/A/M antibody for your ELISA to increase signal detection? Are there concerns for increased background based on prior attempts or experiments done by your or other groups?

(2) Although neutralization was not measured, can you comment on which serological reactivities (particular antigens) would be expected to correlate with neutralization?

(3) The Pearson correlation matrices should be moved to supplemental information unless specific clusters are to be highlighted (see Major Comments).

(4) It is unclear how it was concluded from the PCA that A27 reactivity was the most specific to monkeypox infection to merit that analysis (although it of course worked out). Please explain/clarify further, particularly what analyses were done to determine this.

(5) Please combine Figure 6A and 6B since it is best to show how the pool behaved relative to individual antigens on the same graph, being sure to indicate it clearly (thicker line?).

(6) More of a stylistic comment, but I would maybe consider a title that better highlights your exciting findings such as "Shared yet distinct humoral antigen recognition signatures between mpox-infected and smallpox-vaccinated individuals" or shorter "Humoral antigen recognition signatures between mpox-infected and smallpox-vaccinated individuals."

Reviewer #3:

Remarks to the Author:

In this manuscript by Otter et al., the authors set out to evaluate differences in antibody reactivity against smallpox and monkeypox antigens in individuals vaccinated with smallpox vaccines or infected with monkey pox. The authors identify key antigen targets that are distinct across vaccine and infection cohorts. The manuscript is presented in an approachable manner and the findings are timely. The discussion was very clearly written and nicely framed the findings within the paper. Specific concerns are below.

Major Concern,

1. Can the authors comment on which antibody specificities are associated with protection? How do these responses differ amongst the vaccine and convalescent cohorts.
2. The authors continually only compare the MPOX infected to the Imvanex, but not to the ACAM2000. However, individuals that received the ACAM2000 vaccine generate antibody responses much more similar to MPOX infection than Imvanex. Can the authors explain why this is, despite both vaccines being smallpox? The authors discuss A27 being a culprit, but it is solely responsible for this relationship?

Minor Concerns

1. Line 106 – could you add something regarding synonymous versus nonsynonymous mutations?
2. It's unclear what the axes on Figure 5 and Supplementary Figure 6 are. OD?
3. Line 463 – how long after infection were these samples taken? Could it be taken too acutely that an antibody response would not be detected yet?
4. Paragraph starting at line 584 – Antibody waning is a normal phenomenon of humoral immunity. The real question is whether the waning levels off or continues to decline. It is understandable that these data do not exist for this cohort due to the recentness of this outbreak, but discussion of its relevance is appreciated.

Reviewer #1 (Remarks to the Author):

The main findings of the work by et al. is cross-reactivity between antibody responses directed at monkeypox and vaccinia/smallpox vaccination/infection(s). This finding is not new and dates back to the first descriptions of monkeypox. However, the authors do describe it with the newest strain. The authors also carry out a rather comprehensive serological analysis of monkeypox antigens (less so with vaccinia), and this is the strength of the work. They aim to try to differentiate the responses and find antigen responses or signatures that may help in this.

However, this problem has been tackled before by Mark Slifka (<https://www.ncbi.nlm.nih.gov/pmc/articles/PMC2570942/>) which seemm or claims to have solved this issue. No reference or comparison is given to this work.

We thank the reviewer for highlighting this paper and have added a sentence to our manuscript (lines 595-598) to cover this. However, we believe our differential assay is considerably more robust and easier to perform than listed by Dubois and Slifka (2008). Similarly, the paper highlighted by the reviewer does not identify which specific proteins/antigens could be used as a differential, so we believe the data shown here expands on this knowledge much further as well as further understanding of the immune responses to Mpox infection (both Clades IIa and IIb) and immunological responses to specific poxvirus antigens.

Nonetheless, I feel the work presented in solid and adds substantially to what is known about individual responses to the various proteins, but may be better suited to a more specialized journal.

We appreciate the feedback from reviewer 1 regarding our comprehensive analysis of antibody responses to Mpox infection or Smallpox-vaccination. However, we feel this work is suited and appropriate for publication in Nature Communications as the work described here is applicable to a number of other fields such as serosurveillance, vaccinology, new therapeutic development, and diagnostics

Other minor problems include lack of description of how negative cut-offs determined, as well as functional studies of the antibodies, which the authors point to as well.

This was included within the methods section (lines 215-217), however we have since updated the manuscript to expand on details for negative cut-offs (see line 528).

Reviewer #2 (Remarks to the Author):

Otter et al. performs a comprehensive serological assessment of individuals previously infected with monkeypox or that had received varying doses of two different smallpox virus vaccines (IMVANEX or ACAM2000) and compared them to a large control/confounder cohort to assess serological reactivity (IgG ELISA) to 24 mpox and 3 vaccinia virus antigens at different time points. I believe this is extremely valuable information that will be a great addition to the field both in our understanding of humoral immunity to mpox virus and related vaccines as well as clinical diagnostic test development.

We thank the reviewers for their kind words and suggesting the manuscript is a great addition to the field.

Major Comments:

While I appreciate the importance and quality of this data and the wealth of information presented in this manuscript, it needs to be rewritten to be more concise with less redundancy in data

presentation/figures and clear-cut take away points. It appears that main text figures are repeating the same data in multiple forms of statistical/descriptive analyses that do not add much more value (more specifically, the Pearson correlation matrices and sections of text dedicated to broadly describing correlations). The manuscript main text figures should be limited to the most valuable ones with the most important messages.

For example, in Results, the first three sections should be combined into one, with a title similar to the following: "Mpox convalescent individuals and smallpox vaccinees mount a highly shared yet distinct serological reactivity towards 24 mpox and 3 vaccinia virus antigens." As it is currently written, the first and third section seem to be redundant to each other ("analogous" versus "shared yet distinct") and the second section seems unnecessary. I would use Figure 1 and Figure 3D-E as main text figures for this section. This section would highlight similarities and differences in a general sense, ending with PCA analysis.

We thank the reviewer for highlighting this point. We have combined multiple sections of the manuscript to be more concise and as suggested by the reviewer, titling sections to cover the broader results.

We replaced the original results title from "Antigen recognition is analogous between Smallpox-vaccinated and Monkeypox convalescent individuals" to "Mpox convalescent individuals and smallpox vaccinees mount a highly shared yet distinct serological reactivity towards poxvirus antigens", with removal of subtitles "Correlation is observed between antibody binding to MPXV and VACV homologues" and "Smallpox-vaccinated and Mpox-infected individuals mount a shared yet distinct antibody-binding repertoire".

Further detail has been included within the manuscript regarding an updated Pearson correlation of groups of samples (Fig 2), highlighting that Smallpox-vaccinated and Mpox-infected individuals mount a highly similar antibody response to the different 27 poxvirus antigens. We have then also moved Figure 3A-C to the supplementary material. All correlation matrices shown now utilise the CorrPlot package, with the updated graphs showing only the significant correlations to make it easier to the reader of which correlations are statistically significant and observe which groups correlate to one another.

Then, in Results, sections four and five should also be combined with a title similar to: "Serological reactivity to specific mpox/vaccinia antigens can discriminate mpox convalescents from smallpox vaccinees". This is indeed an important finding to highlight on its own, and is a great follow-up to the PCA analysis that identifies candidates for discrimination of patient groups, first Mpox A27 and Mpox M1 and then Mpox B6 vs.VACV B5 and Mpox A35 vs.VACV A33, which is a great finding. For this I would combine Figures 4 and 5.

We agree with the reviewer regarding these comments and have now renamed one section from "Using MPXV/VACV antigens as a differential for discriminating between MPOX and Smallpox vaccination" to "Serological reactivity to specific MPXV and VACV antigens can discriminate between Mpox convalescent from smallpox vaccinated individuals". We also then removed the subtitle "Antigen exposure influences preferential binding to MPXV or VACV antigens".

Based on the reviewers comments we have now also merged figure 4 and 5 to one figure that summarises the discriminatory nature of individual or combined antigens.

Then, section six should remain on its own as a very nice longitudinal analysis, as well as section seven with the identification of an optimal antigen pool for detection of antibodies in mpox convalescents and smallpox vaccinees.

Again, this is a wonderful paper, but it was difficult to follow and required several re-reads to fully understand specific take-away points. Re-writing/organization would greatly improve the quality to match the high-quality data being presented!

We thank the reviewer for their kind words on our manuscript and we hope that our revised manuscript is much clearer on our results, as we have rewritten multiple sections and reorganised the flow of the manuscript to convey our findings in a more appropriate format from suggestions by the reviewer.

Minor Comments:

(1) Could you comment on the utility (or lack there of) in including IgA and/or IgM antibodies or a pan IgG/A/M antibody for your ELISA to increase signal detection? Are there concerns for increased background based on prior attempts or experiments done by your or other groups?

Yes, this is something we wanted to explore, however, colleagues at the CDC suggested that IgM cross-reactivity is particularly high in poxviruses. So for the work described here, we focussed solely on anti-IgG to ensure limited cross-reactivity. Our current work is expanding into the development of a new IgM assay, but we are also exploring using pan Ig antibody to enable at least some level of broad Ig response as well as IgG specific.

(2) Although neutralization was not measured, can you comment on which serological reactivities (particular antigens) would be expected to correlate with neutralization?

Yes, this is something we are trying to deduce and build on our existing data. One Imvanex dose induces minimal neutralising and binding antibodies, with minimal antibodies able to bind poxvirus antigens B2 and B5, and low level of neutralising antibodies (as described by others). However, two doses induce considerably higher antibody responses, both neutralising and binding, with binding observed to further poxvirus antigens.

Extending from this work, we are conjugating paramagnetic beads with individual Mpox antigens (e.g. B2, B5) and mixing with post-vaccination serum (dose 1 and dose 2) as a mechanism to remove/"deplete" antigen-specific antibodies out of serum. These samples will then be tested for MVA-neutralisation (both normal and "depleted" samples) to determine the impact of removing antigen-specific antibodies on the total neutralising capacity. It is expected that if antibodies to specific antigens are the primary factors involved in neutralisation, we would observe abolishment of neutralisation when depleting samples. However, due to the low number of pox-virus antibodies to specific antigen targets, considerable optimisation is required for this and is ongoing.

(3) The Pearson correlation matrices should be moved to supplemental information unless specific clusters are to be highlighted (see Major Comments).

As discussed in an earlier response to the reviewer, we have moved the individual antigen correlations to the supplementary but kept correlation between groups of samples (e.g., Negatives, vaccinated, Mpox-infected etc) to highlight the high degree of correlation in antibody binding between Mpox-infected and Smallpox-vaccinated individuals, which we feel is an important aspect of this manuscript.

(4) It is unclear how it was concluded from the PCA that A27 reactivity was the most specific to monkeypox infection to merit that analysis (although it of course worked out). Please explain/clarify further, particularly what analyses were done to determine this.

As shown in figure 3B (previously 3E), a biplot is a combination of the PCA analysis and the impact of each variable has on the overall PCA. The Mpx group is skewed towards the yellow grouping, in the bottom right. When using the biplot, the arrows in figure 3B show which antigens result in the biggest variance for the PCA (see below for the overlay of individual variables (black arrows) and colour circles for total groupings) and therefore allowed us to determine which antigens could be used as a differential by performing ROC analysis (table 3). We also observed in the heatmap (figure 1) we also see a Mpx + ACAM2000 specific response.

(5) Please combine Figure 6A and 6B since it is best to show how the pool behaved relative to individual antigens on the same graph, being sure to indicate it clearly (thicker line?).

Yes, we agree with this and have since merged these figures. We have similarly also updated the data in figure 6C post-vaccination endpoint titres up to 185 days post-Imvanex vaccination, providing robust data on antibody waning, further strengthening showing the utility of this assay in measuring antibody responses to infection/vaccination.

(6) More of a stylistic comment, but I would maybe consider a title that better highlights your exciting findings such as “Shared yet distinct humoral antigen recognition signatures between mpox-infected and smallpox-vaccinated individuals” or shorter “Humoral antigen recognition signatures between mpox-infected and smallpox-vaccinated individuals.”

We thank the reviewer for this comment however, we believe our current title is succinct to the work within the manuscript.

Reviewer #3 (Remarks to the Author):

In this manuscript by Otter et al., the authors set out to evaluate differences in antibody reactivity against smallpox and monkeypox antigens in individuals vaccinated with smallpox vaccines or infected with monkey pox. The authors identify key antigen targets that are distinct across vaccine and infection cohorts. The manuscript is presented in an approachable manner and the findings are timely. The discussion was very clearly written and nicely framed the findings within the paper. Specific concerns are below.

We thank the reviewer for their kind comments on our manuscript.

Major Concern,

1. Can the authors comment on which antibody specificities are associated with protection? How do these responses differ amongst the vaccine and convalescent cohorts.

At current, we cannot comment on specific antibody specificities as we have only looked at IgG antibody binding to antigens. However, we are now looking to explore both IgM and IgA levels to antigens in addition to neutralisation titres, from our initial data, it is primarily an IgG response, with IgM observed within 0-7 days post-infection. However, we are limited by these samples, as a number of individuals are PCR positive during the acute phase and thus prevents us with additional working constraints in testing serum harbouring CL3 viruses without validated inactivation methods (and without destroying the antibodies). We also commented to reviewer 2 similarly about how we are also exploring the impact of antibodies to specific antigens on the overall neutralisation titres.

2. The authors continually only compare the MPOX infected to the Imvanex, but not to the ACAM2000. However, individuals that received the ACAM2000 vaccine generate antibody responses much more similar to MPOX infection than Imvanex. Can the authors explain why this is, despite both vaccines being smallpox? The authors discuss A27 being a culprit, but it is solely responsible for this relationship?

The reviewer highlights a very important point and we have since expanded our discussion (lines 634 - 640) to cover this. We show that both Imvanex and ACAM2000 vaccination induces a highly similar antibody response between one another and to Mpox, as observed using the Pearson correlation (Fig 2).

However, the deletion of a number of genes within the Imvanex strain of MVA (e.g. A27) compared to ACAM2000 or Mpox is likely the reasoning for the difference observed in the PCA, as a minimal number of individuals generate a response to A27, driving the differentiation. The similarity between Mpox and ACAM2000 is likely due to the observation of high correlation to all antigens, but also with the shared antibody responses to A27 which is absent in Imvanex.

Minor Concerns

1. Line 106 – could you add something regarding synonymous versus nonsynonymous mutations?

Yes, this has been updated to include synonymous/nonsynonymous mutations

2. It's unclear what the axes on Figure 5 and Supplementary Figure 6 are. OD?

We thank the reviewer for highlighting this – we missed the axes label of OD on these graphs and have since updated these graphs to include this.

3. Line 463 – how long after infection were these samples taken? Could it be taken too acutely that an antibody response would not be detected yet?

We have updated the manuscript (lines 284-285, 636-639) to detail the number of days since their infection for these individuals, as they were 20 and 76 days post-infection and thus should have generated an antibody response. We hypothesise that these individuals had either generated minimal antibody responses to a Mpox infection (possibly coinciding with immunosuppressive diseases such as HIV) or had a false-positive PCR, however, all individuals were symptomatic suggested the former and could be analogous to one dose vaccination in individuals (e.g. low antibody responses)

4. Paragraph starting at line 584 – Antibody waning is a normal phenomenon of humoral immunity. The real question is whether the waning levels off or continues to decline. It is understandable that these data do not exist for this cohort due to the recentness of this outbreak, but discussion of its relevance is appreciated.

We thank the reviewer for highlighting this and agree that waning is of course natural. We have now added a sentence to the manuscript highlighting that this is something we need to measure longitudinally to understand vaccination/infection responses over time.

Author changes unrelated to reviewer comments:

Rectified spelling mistakes

Renamed MPXV/VACV proteins from their gene names (e.g., B2R) to protein names (e.g., B2)

Addition of disclaimer, funding, ethics, data availability and acknowledgements section (lines 948-976)

Rectified error in *Initiative* instead of *Innovations* for CEPI within the acknowledgments (lines 712)

Rectified monkeypox-vaccinated to monkeypox-infected (line 715)

REVIEWERS' COMMENTS

Reviewer #2 (Remarks to the Author):

This is an excellent manuscript that now incorporates significant changes that make it much clearer to read as well represent the data more clearly and concisely. I look forward to seeing the impact it will have in the field both for diagnostic test development and understanding of humoral immune responses to poxviruses.

Reviewer #3 (Remarks to the Author):

While my concerns were relatively minor, the authors have thoughtfully addressed them. I have no further concerns.